# Direct Joule Heating as a Means to Efficiently and Homogeneously Heat Thermoplastic Prepregs

**DOI:** 10.3390/polym12122959

**Published:** 2020-12-11

**Authors:** Jochen Wellekötter, Christian Bonten

**Affiliations:** Institut für Kunststofftechnik, University of Stuttgart, Pfaffenwaldring 32, 70569 Stuttgart, Germany; christian.bonten@ikt.uni-stuttgart.de

**Keywords:** thermoplastic-based composite, injection molding, process monitoring, electrical properties

## Abstract

Although direct Joule heating is a known technique for heating carbon fiber reinforced plastics, it is a yet unexplored heating method for thermoplastic prepregs before back-injection molding. The knowledge obtained from resistance welding, for example, is not directly transferable because of considerably higher heated volumes and more complex shapes. In this study, the governing parameters and process limits are established for this method. The influences of the contacting, the materials used, and the size of the heated part are investigated with respect to the part temperature and heating efficiency. The findings show that the quality of heating is determined by the shape and size of the electrodes. Larger electrodes lead to a more homogeneous temperature distribution. Parts based on woven fabric can be heated more homogeneously because of the existence of intersections between rovings, generating contact between fibers. An increase in part width results in uneven heating behavior.

## 1. Introduction

The force required to move a vehicle at a constant or accelerating speed, determined by the vehicle running resistance equation [1], is comprised of three terms in which mass is a decisive factor: inertia, rolling resistance, and pitch resistance. Accordingly, a reduction in mass is particularly useful in the transport sector. In addition to changes in design and construction, alternative materials can be used alongside steel to reduce mass. Carbon fiber reinforced plastics (CFRP) in particular offer a high lightweight construction potential, but their manufacturing and processing is an ongoing challenge [1,2].

Due to the textile structure of the fibers and the resulting complicated handling, a great deal of manual effort is still required during production [3,4]. Thermoplastic continuous fiber-reinforced parts (thermoplastic prepregs, often also called organic sheets) are used to meet the unresolved challenges of eliminating manual processing and ensuring an efficient and automated production. At the same time, the use of prepregs exploits the well-known advantages of plastic processing, such as the functional integration and reduction of assembly effort, as well as short cycle times [3,4,5,6]. First, the semi-finished thermoplastics are cut to their final shape, then heated using the common heat transfer principles [3], and, after that, transferred into an injection molding tool. Industrial robots are used for automation. Still, there is a high potential for improvement of this comparatively new processing method. In particular, the heating efficiency and handling of the prepregs can be improved.

## 2. Fiber Reinforced Thermoplastics (FRP) and Their Processing

Compared to thermoset FRP, thermoplastic FRP (FRTP) offer several benefits: They are re-meltable, which enables easier repair and further processing, as well as new recycling routes [1,3]. By eliminating long-lasting cross-linking processes, the cycle time can also be reduced [3].

In addition to injection molding, there is a range of further processing methods for FRTP. Most production approaches attempt to minimize the flow of the plastic around the fibers in order to improve impregnation. This can be achieved, for example, by the use of pre-impregnated parts [6,7,8,9,10] or hybrid yarns [11,12]. Other approaches (such as those illustrated in Reference [13,14]) pursue a direct molding approach for previously heated fibers [5].

Established methods for producing pre-impregnated parts are, among others, the powder and solvent impregnation, the textile blending of so-called commingling yarns, the film stacking method, and melt coating [5,15]. A complete impregnation of the fibers is usually achieved by applying pressure using double-belt presses, interval heating presses, or autoclaves. With the high pressing forces and the spatial proximity of fiber and matrix, the limitations of the plastic’s high viscosity can be overcome [5,11]. An overview of the procedure steps and the advantages and disadvantages are listed in Reference [4].

The prepregs are processed afterward via back-injection molding for increased part complexity [3,5,7]. Initial approaches to this combination of processes were investigated in the 1990s [16]. Prominent methods are the in-mold-forming and SpriForm processes, which are both joint projects of part manufacturers, material suppliers, and research institutes. Today, a number of manufacturers offer variants of these methods [17,18,19,20], and parts are produced for serial application [21,22].

### 2.1. Heating of FRTP

The common heating principles, contact heating and infrared radiation, are already used industrially for heating FRTP. Heating in infrared radiation furnaces, in particular, has become established for many applications and is optimized, for example, by the use of several linear robot systems for pick-and-place operations during heating and complex control systems [19,23]. However, the variety of periphery requires an increased investment effort, and the maximum heating speed is limited by a high system inertia and comparatively low efficiency. An efficiency of 40% [24] and maximum heating rates of 10 K/s [25] are achieved for parts with a thickness of 1 mm. The use of contact heaters, on the other hand, has not been adopted widely despite its simple and cost-effective implementation, as the contact can cause damage to the part.

In both contact heating and heating by means of an infrared radiation furnace, heating is achieved via an external energy source. As a result, the parts are heated from the outside; the energy penetrates slowly into the inner core of the part, even with infrared radiation. The heating of the core is mainly generated by heat conduction. This results in a temperature gradient along the part thickness, especially for thick-walled parts, so that temperatures at the surface are too high and the matrix is damaged, while the core is not yet in a shapeable thermoelastic temperature range.

### 2.2. Joule Heating

James Prescott Joule discovered as early as 1850 that an electric current in a conductor is partially converted into thermal energy proportional to the conductor’s electrical resistance [26]. The material reduces the movement of the electrons induced by the electrical potential and thus decreases their kinetic energy. This energy is converted into heat. The principle of Joule heating is used in many applications. Immersion heaters and water boilers, but also seat and steering wheel heaters in automotive applications, are based on this principle. Joule heating is also already used in applications for CFRP, for example, for resistance welding and accelerated consolidation of thermoset matrix systems [27,28,29,30,31]. Especially, resistance welding has matured over the years and is today applied in the aerospace sector for during assembly, as shown by Reference [32]. Usages of Joule Heating for composites processing have been reported by Reference [33] for the direct heating of thermoplastic prepregs and the preheating of carbon fibers before resin injection. More recent investigations, e.g., by Reese, involve the preheating of (recycled) carbon fiber hybrid yarn [34]. The dry fibers can then be processed and consolidated in short cycle times, as shown in Reference [35]. Additionally, Joule heating is widely investigated in the comparatively new field of carbon nanotubes for, e.g., production of carbon nanotubes [36] or faster curing of composites [37].

In the context of this study, the principle of Joule heating is used to heat CFRTP for back-injection molding. In contrast to existing usages of Joule heating, especially resistance welding, this is a new field with little research, different requirements, and has to overcome different hurdles. To the author’s knowledge, investigations have thus far only been conducted by Hemmen [33]. Parts for back-injection molding are usually two-dimensional and need to be heated homogeneously over the whole plane of the part, instead of only at the weld line. This is particularly challenging when utilizing Joule heating because of a varying current density, especially for complex part shapes.

However, Joule heating offers high potential for more efficient production. The heating process can be integrated into a robot handling system, thus avoiding transport into an infrared radiation furnace and shortening the heating process. In contrast to most other heating methods, a part is heated from the inside over the entire volume by Joule heating [33]. On the one hand, this means that the part can theoretically be heated very quickly and uniformly and, on the other hand, that a high efficiency of theoretically more than 90% [38] can be achieved, since only minor losses occur due to convection and heat conduction. The results of various basic investigations on resistance heating have already been extensively published by Koslowski and others [33,38,39,40,41]. The authors know of no further applications of Joule heating for back-injection molding.

### 2.3. Aim

The effects of Joule heating on resistance welding have been extensively analyzed but have not been applied to planar parts with higher surface areas. Therefore, the hypothesis is derived that the use of Joule heating of carbon fiber reinforced thermoplastics can be extended from resistance welding to heating of prepregs for back-injection molding, and that sufficiently fast and efficient heating can be achieved while maintaining a narrow temperature distribution.

In this study, the governing processing parameters are therefore systematically analyzed. This study focuses on the influence of the contacting of the part on the resulting heating rate and temperature distribution. We assume, that with a suitable choice of contacting parameters, a heating efficiency close to the theoretically possible efficiency can be achieved. Additional experiments show the influence of the materials used and the part’s size. The material is expected to have a high influence on the heating behavior since preliminary experiments on unidirectional fiber tapes showed less homogeneous heating than woven fabrics. A change in the part’s size, on the other hand, is expected to require more energy input and thus less homogeneous heating due to higher demands on the contacting.

## 3. Experimental Setup

The aim of this work is to systematically analyze the Joule heating of thermoplastic prepregs. In order to determine the influencing parameters, a test stand is developed on grounds of the basic investigations carried out by the authors and other groups (see Reference [38,39,40]). The test stand enables the monitoring of various process parameters. The contact force, the material used for the electrodes, and the electrode surface, as well as the current and voltage, can be varied. A thermography system VarioCAM^®^ HDx head 615S from InfraTec, Dresden, Germany, is used for temperature data acquisition. The thermography system is calibrated to thermoplastic prepregs in a range from 0 to 600 °C to guarantee accurate temperature measurements over the prepreg’s surface. Thermocouples can thus be omitted, and the overall surface area of the heated part can be analyzed. The thermography system communicates with a computer via an Ethernet interface and transfers the recorded images in the form of temperature arrays (matrix with a temperature value per pixel) directly via the application programming interface (API) of the measuring program to the graphical programming system LabVIEW^®^ by National Instruments, Austin, TX, USA.

A power supply PS9000 3U of EA Elektro-Automatik GmbH, Viersen, Germany, supplies output currents between 0 A and 340 A and provides output voltages between 0 V and 80 V at a maximum power of 15 kW. The power supply has an internal automated control. The parameter whose threshold value is reached first is always used as the control parameter. The other parameters result automatically. This will be explained briefly using an example: the control parameters are set to 16 A for the current, 50 V for the voltage, and 1000 W for the power. The overall resistance of the circuit determines which parameter is the control parameter. If the current is 16 A and the voltage of 20 V is reached, the resulting power can be determined as 320 W, and the current is set as the control parameter. If the current is 8 A when a voltage of 50 V is reached, equaling 400 W, the voltage is set as the control parameter because its limit is reached first. With respect to this test series, this indicates that comparatively high voltages should be set and the actual voltage should be determined over the total resistance of the system $R_{total}$. The total energy consumption of the heating process can then be determined.

It should be noted that the current is always set as the control parameter to minimize the effect of the contact resistance on the heating of the part; Joule heating occurs due to the movement of electrodes in a conductor (current), not their potential energy (voltage).

The contact force is applied by a vise. The contact force can be monitored manually using a calibrated hydraulic load cell WASA of AWB Apparatebau GmbH, Osterode, Germany. The electrodes are connected to the power supply and fixed to the vise so that the ends of the part can be contacted. Figure 1 shows a schematic of the test stand including the thermography system.

Unless otherwise stated, thermoplastic prepregs of type TEPEX dynalite 201C200/50% manufactured by Bond Laminates GmbH, Brilon, Germany, in 1-mm thicknesses, are used for the experiments. The industry-grade thermoplastic prepregs are used in series production for a plethora of automotive parts, for example, and are well established. Heating is usually achieved via infrared furnaces. This allows for a direct comparison of Joule heating with established methods in an industrial setup. The prepregs are based on a carbon fiber fabric in 2/2 body weave and 3K-rovings consisting of 3000 individual filaments. A single fabric layer has a thickness of 0.25 mm and weighs 200 g/m^2^. For the matrix, a polyamide 6.6 (PA66) is used. The material data sheet can be found in Reference [42]. All prepregs are stored for at least 100 h without conditioning. Preliminary tests did not show any relation between conditioning and heating behavior.

It is assumed that the prepreg’s fiber structure and fiber volume content have a strong influence on the heating behavior. However, due to the availability of different prepreg materials, these parameters cannot be investigated in this study. Changing the fiber structure always requires prepregs of a different manufacturer, which also changes the manufacturing process, the matrix materials used, and other parameters. The authors are well aware that this limits the informative value of the results, which is why an additional test series with varying materials is carried out (see Section 3.1.2).

Data entry and acquisition is achieved via the programming environment LabVIEW^®^. The emission coefficient ε is assumed to be 0.9. Investigations by the authors have shown that the assumed coefficient is sufficiently accurate within the range of experimental accuracy, especially since the goal is to reach homogeneous heating and absolute temperatures are of minor importance in this study. The temperature arrays from the thermography system and the test data are saved as text files in the so-called tidy-data format [43]. All heating tests are carried out on two different parts and repeated two times, resulting in four test repetitions. The evaluation of the average part temperature and temperature distribution is carried out after 10 s heating time in order to achieve comparable testing conditions. This testing method allows for the comparison of a wider range of parameters: using target temperature as a test condition would result in test failures if the temperature is not reached and corrupted data if the temperature is reached too quickly.

### 3.1. Test Series

A part’s size and shape define the energy required to heat it. This affects the current throughput at the electrodes and therefore is assumed to have an influence on the heating efficiency and the temperature distribution after heating. For the analysis of the influence of the contacting and part’s size on the heating behavior, statistical analysis methods according to the design of experiments (DoE) [44,45] are used. However, for the analysis of the influence of the material on the heating behavior, this is not feasible due to availability of material combinations. Instead, direct comparison is carried out in this case. The test series conducted in this study are based on previous findings. See [38,39,40,41] for further details and the range of parameter variation. Additionally, the authors conducted a parameter study beforehand using the FEM simulation software Comsol Multiphysics^®^, Comsol Inc., Stockholm, Sweden, in order to further narrow down the range of process parameters. A detailed view of the simulation parameters can be found in Reference [46].

#### 3.1.1. Influence of the Contacting on the Heating Behavior

Tests are carried out in a full factorial design according to Table 1. The influencing factors current *I*, contact force *F*, contact width *b*, and the contact shape *O* are analyzed. During all experiments, copper electrodes are used. According to Reference [33], the contact shapes are designed with and without rounded edges as seen in Figure 2. The reduction of effective width due to rounded edges needs to be taken into account during an evaluation of the results. The parts are cut into bar-shapes of 150 mm length and 25 mm width, according to ISO 527-4 [47].

#### 3.1.2. Influence of Textile Structure and Matrix Material on the Heating
Behavior

Thermoplastic prepregs usually differ considerably from manufacturer to manufacturer. A comparison of different textile materials, matrix materials, and manufacturers should provide further insights into the potential of Joule heating to illustrate universal transferability. The underlying effects of Joule heating are known due to the research done on resistance welding and preliminary tests that showed that fast heating is possible. Therefore, industrial grade prepregs are analyzed to gain insights for processing in an industrial setup. A direct comparison of the fiber content, for example, which should be the most influencing factor on heating, cannot be presented in this study, however, since industrial grade prepregs usually do not come with varying fiber content; the fiber content depends on the manufacturing method. Table 2 lists the materials that are used for these experiments. Industrial grade prepregs from three different manufacturers are used. The prepregs from Bond Laminates used in the first series of experiments are also used here and compared to specifically manufactured tape-based parts from SGL Carbon SE, Wiesbaden, Germany, and parts from Ineos Styrolution GmbH, Frankfurt, Germany, of type StyLight^®^ Aesthetic. The first is chosen to compare the results of a woven-based prepreg to one manufactured by tape layup; the latter is chosen to compare two different woven-based prepregs with a different matrix material.

The influences of different matrix materials and textile structures on heating behavior are determined. All prepregs have a comparable fiber volume content of 45 to 50%. Two different textile structures are analyzed: parts based on woven fabrics and parts based on tape layup. The layer structure of the tape-based prepregs is [0; 90; +45; −45; 90; 0]. As a result of the first test series, the current is set to 16 and 18 A, respectively.

The results are illustrated in boxplots. The boxplots contain the mean value of the data (indicated by a dot), the median (marked by a horizontal line), as well as the quartiles, marked by the box. The distribution function (kernel density estimation) is also displayed. All values are tested for significant differences and the significance level is calculated using the analysis of variance (ANOVA) [48].

#### 3.1.3. Influence of the Part’s Size on Heating Behavior

Scaling of the heating to larger parts is an important quality criterion for industrial use. Therefore, the influence of the part size on the Joule heating has to be investigated. It can be expected that scaling introduces new effects due to Joule heating compared to previous research on resistance welding. Preliminary tests showed, for example, that much higher contact forces are required to provide an efficient current flow into the part without hot spot formation.

The parts are scaled on a laboratory scale according to Table 3. The part’s width and length are varied. Table 3 also shows the designations used for the factors examined. The parts used are identical to those used in the investigation of the influence of contacting. Since the shape of the prepregs is not changed, the current flow paths should not change compared to smaller part sizes.

It should be noted in particular that doubling one part’s dimensions leads to a doubling of the part’s volume, i.e., twice the energy is required for heating. At the same time, when contacting in a longitudinal direction, doubling the length can be understood as a series connection of two resistors, but doubling the width can be understood as a parallel connection (the equation for calculating the resistance of a wire also applies here; see Equation (Equation 1)).
(1)$$R=\rho_{el}\frac{L}{A},$$
with $\rho_{el}$ as the specific electrical resistance of the material, and *L* and *A* the length and cross section of the resistor, respectively.

This means that the set current should remain constant when the length is changed but must be adjusted to the same extent when the width is changed in order to obtain comparable settings and an equal heat input. For 50-mm wide parts, a current of 32 A or 36 A is set accordingly, resulting in the same current density for those parts.

### 3.2. Evaluation Algorithms and Methods

The evaluation of the test data is performed as described in Reference [40], by first using a filter algorithm that isolates the heated part from the environment. The method of Savitzky and Golay [49] is used to smooth local temperature fluctuations. The ambient temperatures are masked and a limit gradient value is defined, which corresponds to the minimum temperature of the part. On the basis of the isolated part, the *average part temperature*
$\vartheta_{avg}$ (the mean value of the temperature values of all pixels of the part; see Equation (Equation 2)), as well as the *temperature distribution*
σ (relative standard deviation of the temperature values of all pixels of the parts; see Equation (Equation 3)) can be determined and used for further evaluation.
(2)$$\vartheta_{avg}=\frac{\sum \vartheta_{Pixel}}{n_{Pixel}},$$
(3)$$\sigma =\frac{\sqrt{\frac{\sum {\left(\vartheta_{Pixel}-\vartheta_{avg}\right)}^{2}}{n_{Pixel}}}}{\vartheta_{avg}},$$
with $\vartheta_{Pixel}$ and $n_{Pixel}$ the temperature per Pixel and the number of Pixels recorded by the camera, respectively.

The *efficiency*
φ is composed of the ratio of the theoretically required energy ($Q_{required}$) to heat the part from ambient temperature ($\vartheta_{0}$) to the achieved *average part temperature*, while considering the loss of heat ($Q_{losses}$) and the actual consumed energy ($Q_{used}$).
(4)$$\begin{array}{c} \varphi =\frac{Q_{required}+Q_{losses}}{Q_{used}}. \end{array}$$


It is estimated that the temperature curve is approximately linear over time for simplicity, and the heat transfer coefficient has a constant value of 10 W/m^2^K. This value, often used to display losses in transition and radiation, represents a rather conservative estimate and is below the values specified in ISO 6946 [50]. The part’s specific heat capacity is determined by means of differential scanning calorimetry (DSC). When calculating the area of the part, the edges are not considered, since this area is obscured by the electrodes. The resulting *efficiency* can be calculated according to Equation (Equation 5):
(5)$$\begin{array}{c} \varphi =\frac{\rho_{part}V_{part}\int_{T_{0}}^{T}c_{p}dT-A\int_{}^{t}\int_{T_{0}}^{T}\alpha dTdt}{\int_{}^{t}UIdt}, \end{array}$$
where $\rho_{part}$ is the density of the material, $V_{part}$ the volume of the part, $c_{p}$ the specific heat capacity, *A* the part’s surface without the edges, and α the heat transfer coefficient. *U* and *I* are the voltage and current, respectively.

## 4. Results

The heating of a bar-shaped part over a period of 20 s and its corresponding temperature distribution are illustrated in Figure 3 as a function of the current. Wide electrodes (12 mm) with rounded edges and a contact force of 20 kN are applied. As is expected from previous research on resistance welding, heating takes place. The current flow path over the entire part allows for a homogeneous heating of the entire volume.

Figure 4 shows the temperature distribution of the same part after 10 seconds heating time at 16 A. As can be seen from the narrow temperature distribution, a uniform current flow path through the part can be expected with higher convective cooling in the boundary areas. In addition, the underlying structure of the woven fabric can be seen, demonstrating the heating progress from fiber to matrix via heat conduction. Due to heating, delamination of the woven layers occurs. This has also been observed by Hemmen [33]; however, since an injection molding process is started afterwards, this effect can be neglected and is common also for parts heated in with infrared radiation.

As can be seen in Figure 3, because of small differences to the ambient temperature, it is not possible to record the part temperature in the first seconds. Once the part temperature sufficiently differs from the ambient temperature, the recording starts. The temperature rises degressively over time, starting at room temperature and moving towards a limit value. As expected, a higher current causes faster heating. The limit is reached when the heat flows from free convection and heat radiation reaches the same value as the supplied energy from Joule heating. Temperature-dependent changes in the part’s specific resistance lead to changes in the supplied heat flow over time, which is why even without heat losses a non-linear but progressive course occurs. Overall, the heating of the part occurs as expected.

Homogeneous heating leads to a constant *temperature distribution* over the entire heating period for a current of 16 A and 18 A. Compared to resistance welding where only a small volume needs to be heated, a small distribution is even more important to achieve. It should be noted that the fibers heat up due to Joule heating, and the heat is distributed in the matrix by means of heat conduction. The measured temperature at the part’s surface is therefore delayed by this effect. This is especially true since part dimensions are considerably higher compared to welding areas. Still, a relatively small temperature distribution of up to 6% can be observed, suggesting an overall homogeneous heating and a high heating quality. The current flow path appears to be uniform along the part.

### 4.1. Influence of the Contacting on the Heating Behavior

The test series investigating the influence of the contacting on the heating behavior is primarily conducted to optimize the process window of the Joule heating process. Since surface area and volume of the heated part significantly differ from resistance welding, preliminary tests were conducted in Reference [39,40] to give closer insight into the range of the parameters that needed to be set.

Figure 5 shows the *average part temperature* and the *temperature distribution* of all tests in this series of experiments after 10 s heating time. The temperatures reached range between approximately 150 °C and 230 °C, resulting in heating rates of 13 K/s and 21 K/s, respectively. This is well beyond the maximum heating rates of 10 K/s [25] for infrared radiation. Koslowski [38] and Geyer [39] recorded heating rates of even more than 60 K/s without considerable influence on the mechanical properties of the materials. However, homogeneity has not been recorded in these cases.

The *temperature distribution* (relative standard deviation, related to the average part temperature) varies between 2 and 7% and can be considered comparatively low. Figure 5 also provides an initial reference on the effect of varying currents on the *average part temperature* and the *temperature distribution*. Tests with a current of 16 A achieve significantly lower temperatures than the tests at 18 A, which is in line with the displayed temperature curve. However, the temperature distribution is observably smaller for the tests at 18 A.

Figure 6 asserts the influence of the investigated factors on the *average part temperature*. The effects of the evaluated parameters are shown and the statistical significance of these effects according to a p-test by ANOVA is given. If an effect ist statistically significant, the difference of the average temperature is sufficiently large to not be a result of random scattering of the results. The effect size is illustrated by the slope of the graph of each parameter. A higher change in average temperature indicates a higher effect of the factor on the average temperature. An effect of $E_{\vartheta_{avg},I}$ = 32.73 is determined for the current and can be classified as highly significant. This implies that an increase in the current of 1 A results in an increase of the *average part temperature* by roughly 16 K after ten seconds heating time.

The shape of the electrodes and the contact force do not influence the *average part temperature*. The electrode width, however, has a significant effect $E_{\vartheta_{avg},b}$ = −16.2. Therefore, an increased contact width results in a reduction of the *average part temperature*. This effect will be discussed in more detail below, in connection with the effect of the contact width on the *temperature distribution*

Figure 7 illustrates the effects of the factors on the *temperature distribution*. A significant effect $E_{\sigma ,I}$ = −0.5 of the current on the *temperature distribution* can be determined. An increase of the current results in a reduction of the *temperature distribution*, which ensures more homogeneous heating. This allows room for the assumption that with an increased current, the current flow through the part is more evenly distributed. Cavities and irregularities in the contact surface can be overcome more easily at higher currents.

According to Reference [30], in resistance welding, the contact resistance from the electrode to the part decreases with increasing contact force, resulting in a more homogeneous transmission of the current. This can, however, not be confirmed in this study; no influence of the contact force on the *temperature distribution* can be determined. Since the applied contact forces are substantially higher than those investigated by Reference [30], it is assumed that after a limit is reached, no significant improvement with regard to contact resistance can be detected. In addition, the overall volume to be heated is higher compared to resistance welding. Therefore, higher contact forces are required, as could be observed in preliminary tests by the authors [40].

In contrast to the results for the *average part temperature*, a slightly significant effect $E_{\sigma ,O}$ of −0.35 can also be determined for the contact shape. Rounded edges appear to ensure a gentler flow, preventing the formation of hot spots. This effect is reinforced by the fact that rounded edges lead to a reduction of the effective contact width since the effect of the electrode width with a value of $E_{\sigma ,b}$ = −1.02 is categorized as highly significant. By increasing the width of the electrodes, a more homogeneous *temperature distribution* is achieved. Since the rounded edges reduce the contact width but have a positive effect, it can be assumed that the positive effect is greater if the electrode shape remains unchanged. The overall energy input is not dependent on the electrode configuration (see Section 3), but the temperature distribution (hot spots) is.

With increasing contact width, the irregularities in the *temperature distribution* are reduced since hot spots, as well as sparks and small burns, can be avoided. This also leads to a slight reduction of the *average part temperature*. As shown in Figure 8, the temperature distribution does not correspond to an ideal Gaussian distribution but is shifted towards lower temperatures. Therefore, the temperature distribution is sensitive to the influence of hot spots.

The heating *efficiency* is shown in Figure 9. All factors except the contact shape have a significant effect on the *efficiency*. The effect of the current ($E_{\varphi ,I}$ = 0.02) corresponds to an improvement in efficiency of 1% per ampere. The effects of the contact width and the contact pressure are $E_{\varphi ,b}$ = 0.04 and $E_{\varphi ,F}$ = 0.03, respectively.

Apparently, the contact resistance has a decisive influence on the heating *efficiency* since lower contact resistance, because of wider electrodes and higher contact forces, reduces the current losses through the part. Thus, more power is available for heating the part. The *efficiency* is calculated with an average of 70% and a standard deviation of 6% and is regarded as extremely high.

In summary, it can be concluded that higher currents, larger contact areas, and rounded electrode edges lead to significantly better heating results. Heating of the entire volume of the thermoplastic prepreg is possible for all parameter settings. The resulting *average part temperature*, *temperature distribution* and *efficiency* are summarized in tabular form in Table 4. The significant effects are highlighted in green.

### 4.2. Influence of the Material on Heating Behavior

In this series of experiments, the influence of the material on the Joule heating of thermoplastic prepregs is evaluated. In addition to testing different matrix materials, different parts from different manufacturers are examined. A design of experiments cannot be carried out due to the varying availability of matrix and textile combinations. Therefore, a comparison using boxplots is defined for the evaluation of the results. The probability distribution is also given in each case.

The *average part temperature* and the *temperature distribution* of all tests of this test series after 10 s heating time is illustrated in Figure 10. Scattering of the resulting *average part temperature* is comparatively low. The *temperature distribution*, however, is considerably higher compared to the previous test series. This applies in particular to tape-based parts. The influence of the textile structure on the *average part temperature* and the *temperature distribution* is given in Figure 11 and Figure 12, respectively.

For both woven fabric and tape-based parts, an *average part temperature* of approximately 180 °C is reached after 10 s heating time. This is in accordance with the results of the first series of experiments. The significance level of 0.99 shows that the mean value can statistically not be distinguished. The textile structure has no measurable influence on the *average part temperature*. This would be represented by an Effect $E_{\vartheta }=0$ if a DoE was conducted.

However, a significant difference in the *temperature distribution* for woven fabrics is visible. This results in considerably lower *temperature distributions* for fabrics and thus a more homogeneous heating of the parts. For woven fabric-based parts, a large deviation between the median and the mean value of the distribution can be observed. The distribution is highly influenced by only a few measurements with relatively high temperature deviations because of hot spots, for example, while the majority of the measurements show only a small deviation. This effect is due to the part’s textile structure: Tape-based parts are built layer by layer with only a few intersections between fibers. If the contacting of the part is uneven, only a few fibers will actually contribute to the flow of current and heat up with extreme current densities while the rest of the part stays almost at room temperature. Woven fabric-based parts, on the other hand, have a high number of intersections, resulting in a more homogeneous current distribution and thus more homogeneous heating. This effect is illustrated in Figure 13 with a schematic and a corresponding microscopic image of the textile structures.

The kernel density estimation of the *average part temperature*, shown in Figure 11, displays a bimodal distribution. This is a result of the different matrix materials as a combination of individual Gaussian distribution results and the respective average temperatures. The bimodal distribution of measured values represents an overlay of the effects of the matrix materials. Figure 14 and Figure 15 yield further information on the *average part temperatures* and the *temperature distribution* for the matrix materials PA6, PA66, PS, and PP.

On the basis of the results presented, it can be seen that *average part temperatures* vary only slightly. The difference is nevertheless statistically significant, so the heating rate is generally dependent on the matrix material used. This is due to the different specific heat capacities of the materials. If a material requires a larger amount of energy to be heated to higher temperatures, this energy must be provided by Joule heating. Of the four materials investigated, PA66 and PP have the highest *average part temperatures* and, according to Reference [51], the lowest specific heat capacity.

The influence of the matrix material on the *temperature distribution* also turns out to be significant. Especially for the matrix materials PS and PP, an inhomogeneous heating process can be determined. However, the results show a broad scattering for all examined matrix materials. The repeatability of experiments in this test series is therefore significantly lower than in the previous test series.

In Figure 16 and Figure 17, the influence of the matrix materials on the *temperature distribution* is shown for woven fabric-based and tape- based parts, respectively. Accordingly, PS shows an inhomogeneous *temperature distribution* in the case of woven fabric-based parts. On one hand, this can actually be caused by the matrix. On the other hand, these parts were acquired from different manufacturers. Therefore, the manufacturing process of the parts can also play a decisive role. It is, for example, possible that the manufacturing process of this manufacturer results in fewer fibers near the part’s surface, resulting in higher contact resistance.

However, this cannot apply to the non-woven parts, as both matrix materials were acquired by the same manufacturer and produced with the same manufacturing process. Nevertheless, a different *temperature distribution* is observed: PP shows a less favorable heating behavior. This behavior appears to be a result of the material’s electrical conductivity, as shown in Figure 18: Polyamide shows a significantly lower electrical resistance compared to polypropylene. Since all tests are carried out at room temperature and without prior conditioning of the test specimens, it can be concluded that the ability of polyamide to absorb moisture improves electrical conductivity and reduces contact resistance, resulting in an improved *temperature distribution*.

In summary, the Joule heating of CFRTP appears to be only feasible for woven fabric-based parts. Tape-based parts are heated unevenly because of the low number of intersections between fibers and therefore cannot achieve good process quality. The influence of the matrix materials, however, does not show any obvious differences. Due to the limited availability of material combinations, though, these results cannot be further determined. Nevertheless, polyamide appears to be the most suitable material for Joule heating, due to its comparatively high electrical conductivity.

### 4.3. Influence of the Part’s Size on the Heating Behavior

For a successful industrial application of Joule heating, scalability to parts of different sizes is required. In this series of experiments, the influence of size on the heating behavior is therefore investigated. The part width and length are varied systematically in a full factorial design.

Figure 19 shows the *average part temperature* and the *temperature distribution* of all parts after 10 s heating time. In principle, a similar course as in Figure 5 can be determined, even though higher temperatures are achieved overall with a larger standard deviation. The heating of larger parts therefore obviously turns out to be more difficult.

The effects of the influencing factors part length and part width on the *average part temperature* are shown quantitatively in Figure 20. Figure 21 shows the corresponding effects on the *temperature distribution*. Again, the effects are given by the slope of the curves and the significance according to a p-test by ANOVA are given in the graphs. The effects indicate, by how much the target value changes because of factor changes.

For the *average part temperature*, a significant effect ($E_{\vartheta_{avg},J}$ = 31.16) can be determined for the current density only. This result is in line with the findings from the series of tests to determine the influence of contacting. However, the current density has no significant influence on the *temperature distribution*. Here, changes in the part length have a significant effect $E_{\sigma ,J}$ of −0.55. An increase in the part length thus leads to a significantly more homogeneous heating. This is due to the fact that the heating is basically uniform and only a few small hot spots, especially in the contact area, influence the *temperature distribution*. For longer parts, these hot spots are less significant compared to the larger surface of the part, leading to a lower *temperature distribution*.

In summary, it can be stated that with regard to the influence of the part size on the heating behavior of thermoplastic prepregs by means of Joule heating, the part length plays hardly any role for fast and homogeneous heating. The part width, on the other hand, leads to significantly worse results in comparison. This can possibly be remedied by using several independent control circuits over the part’s width, thus guaranteeing a homogeneous current density. The results are summarized in tabular form in Table 5. The significant effects are highlighted in green.

## 5. Conclusions

In theory, Joule heating of fiber-reinforced plastics has high potential. The method shows promising results in many applications. This study proves that the method can also successfully be applied to heat CFRTP before back injection molding. The governing process parameters are provided and analyzed. The process limits applicable to this process in terms of contacting, the materials used, and the part’s size were investigated. Special attention was paid to an energy efficient, fast, and homogeneous heating process.

It became apparent that the quality of the contacting is mainly determined by the shape and size of the electrodes. Here, large electrodes should be used in order to achieve a more homogeneous temperature distribution. The temperature distribution can be further improved by rounding the edges of the electrodes.

The influence of the matrix on the heating behavior can be regarded as low. Only the specific heat capacity of the matrix materials changes the average part temperature according to the energy absorption of the materials. In contrast, the textile structure has a clear influence on the heating behavior. Parts based on woven fabric can be heated more homogeneously due to the many interloops between the rovings.

Scaling of the part’s size proves to be fundamentally feasible. The change of a part’s length is particularly suitable; this corresponds to a series connection of resistors. For a change in part width, however, it should always be considered whether several independent circuits could lead to a more homogeneous heating behavior.

In principle, the knowledge gained in this study can be used to describe the process limits of Joule heating of thermoplastic prepregs much more precisely. Nevertheless, there is still a need for further research. Among other things, further materials should be investigated, for example, with matrix systems for high-temperature applications in the aerospace industry. Since only industrial grade materials could be used in this study, a further exploration of the material’s influence should consider fiber content and fiber orientation, which are expected to have a major influence on the current flow path and, therefore, on heating quality. Furthermore, it is still necessary to transfer the results obtained here to an industrial environment. For this purpose, the parts can be contacted in the end effector of an industrial robot. This robot can heat the parts during handling, thus avoiding transport into the infrared radiation furnace and shortening the heating process.

## Figures and Tables

**Figure 1 polymers-12-02959-f001:**
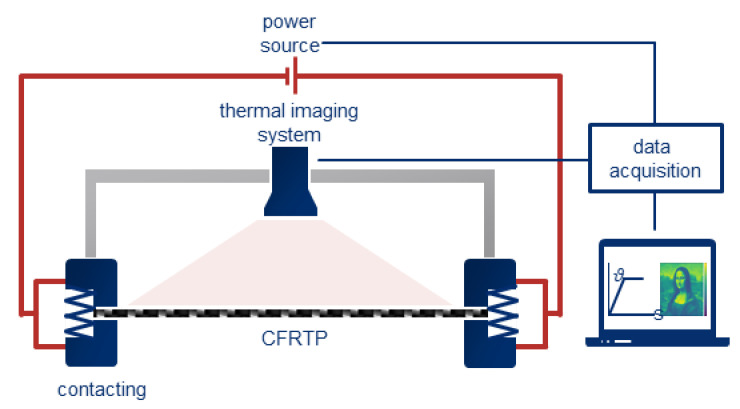
Schematic of the test stand for the Joule heating of thermoplastic prepregs; the contact force is applied using a vise.

**Figure 2 polymers-12-02959-f002:**
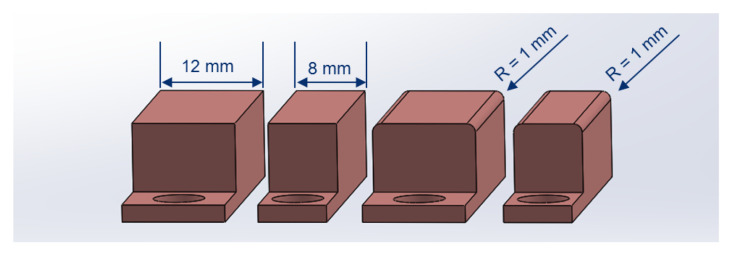
Electrodes used for the contacting of the part.

**Figure 3 polymers-12-02959-f003:**
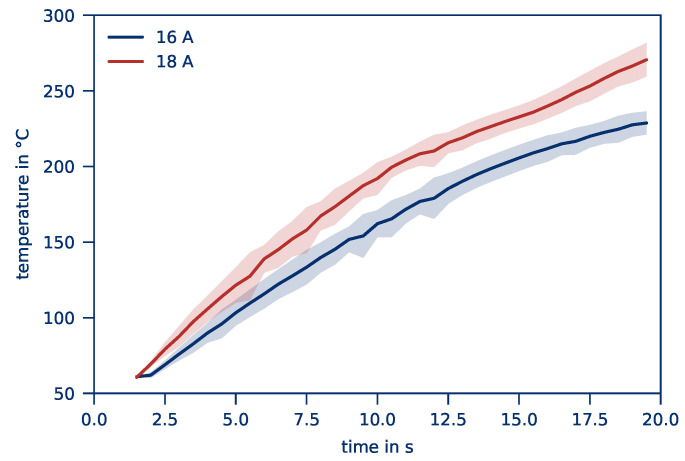
*Average part temperature* and *temperature distribution* depending on the set current of a part heated for 20 s with 12 mm contact width, rounded electrodes, and 20 kN contact force.

**Figure 4 polymers-12-02959-f004:**
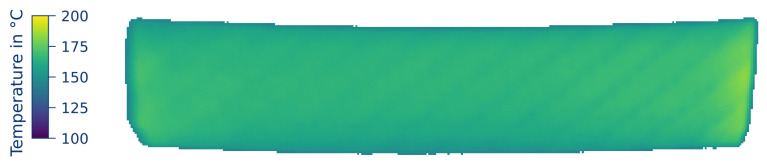
*Temperature distribution* of a prepregs heated for 10 s at 16 A with 12 mm contact width, rounded electrodes, and 20 kN contact force.

**Figure 5 polymers-12-02959-f005:**
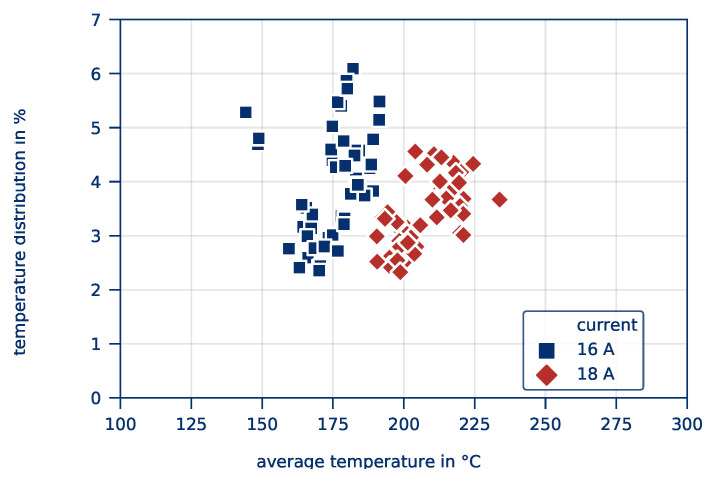
*Average part temperature* und *temperature distribution* as a result of varying currents for all experiments of the test on the influence of the contacting.

**Figure 6 polymers-12-02959-f006:**
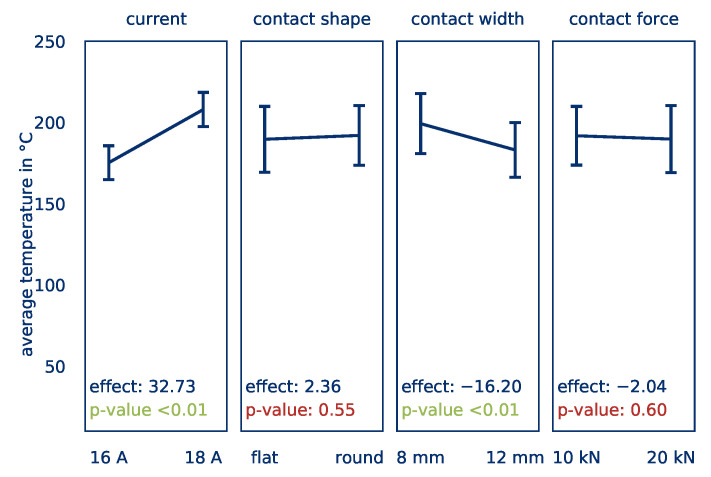
Calculated effects of the current, contact shape, contact width, and contact force on the *average part temperature*.

**Figure 7 polymers-12-02959-f007:**
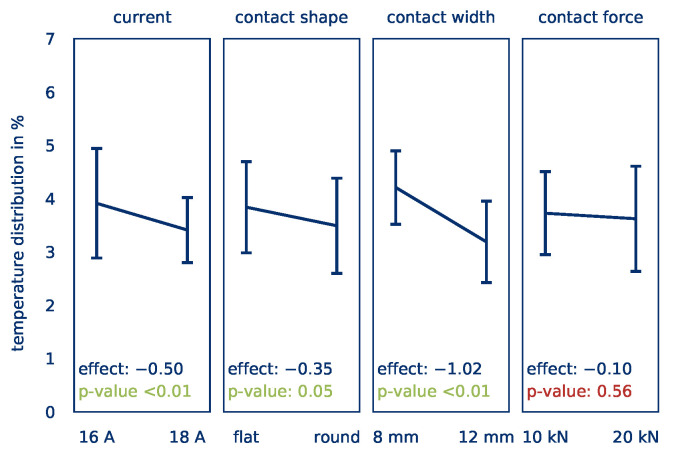
Calculated effects of the current, contact shape, contact width, and contact force on the *temperature distribution*.

**Figure 8 polymers-12-02959-f008:**
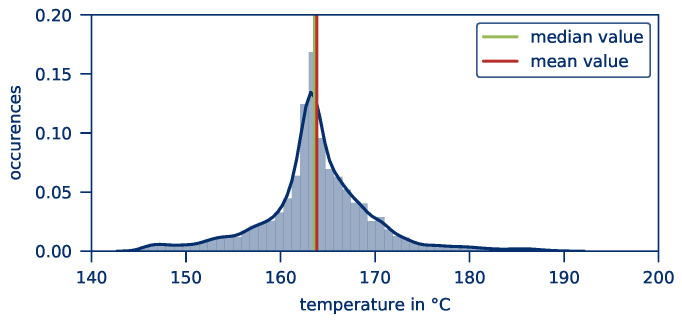
Kernel density estimation of the temperature after 10 s heating time.

**Figure 9 polymers-12-02959-f009:**
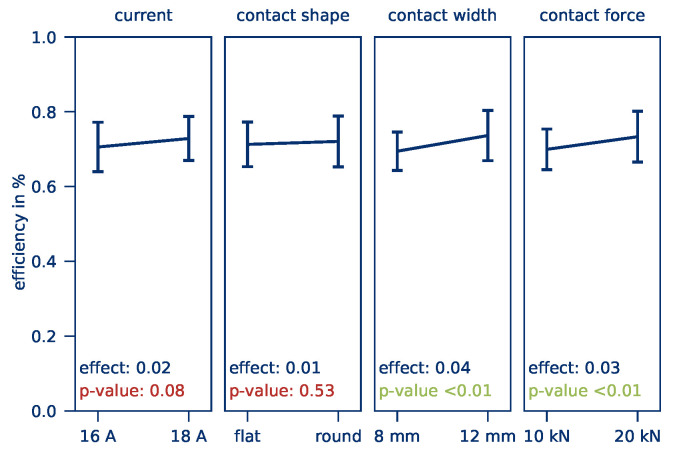
Calculated effects of the current, contact shape, contact width, and contact force on the *efficiency*.

**Figure 10 polymers-12-02959-f010:**
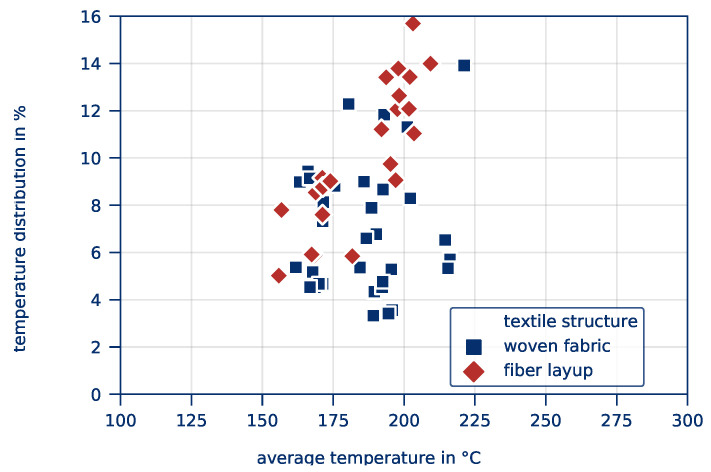
*Average part temperature* and *temperature distribution* as a result of varying textile structures for all experiments of this test series.

**Figure 11 polymers-12-02959-f011:**
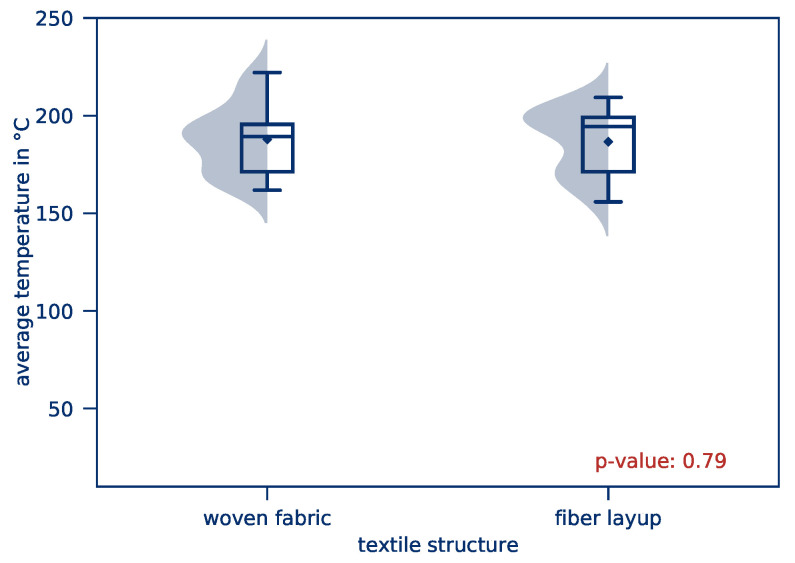
*Average part temperature* depending on the textile structure.

**Figure 12 polymers-12-02959-f012:**
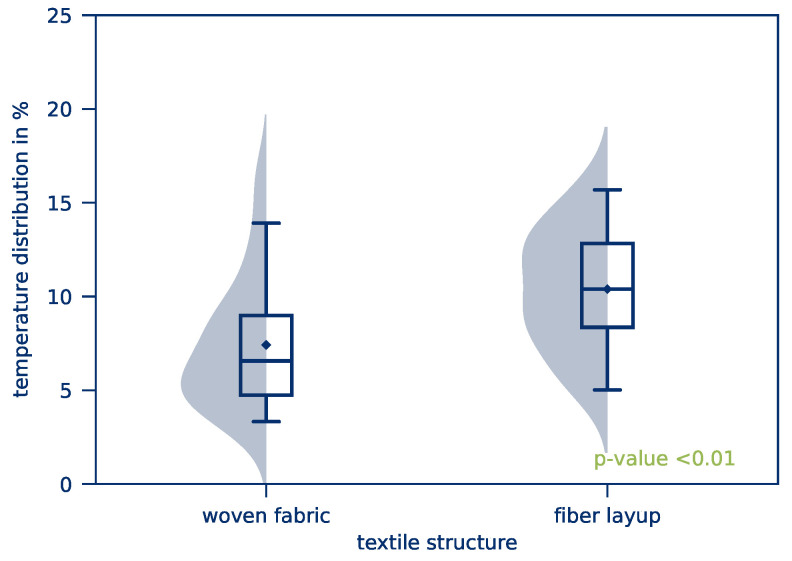
*Temperature distribution* depending on the textile structure.

**Figure 13 polymers-12-02959-f013:**
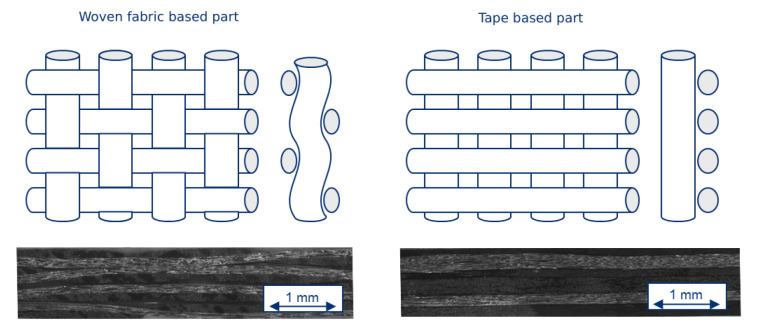
Schematic and microscopic images of the different textile structures.

**Figure 14 polymers-12-02959-f014:**
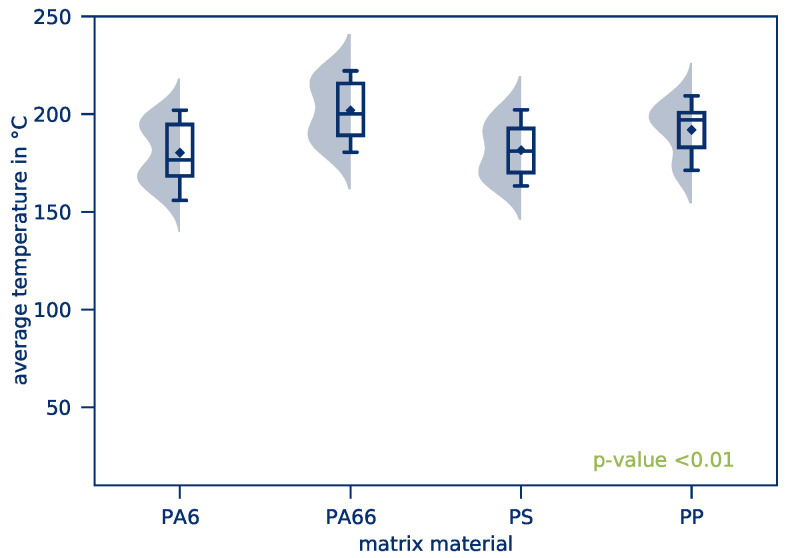
*Average part temperature* as a function of the matrix material.

**Figure 15 polymers-12-02959-f015:**
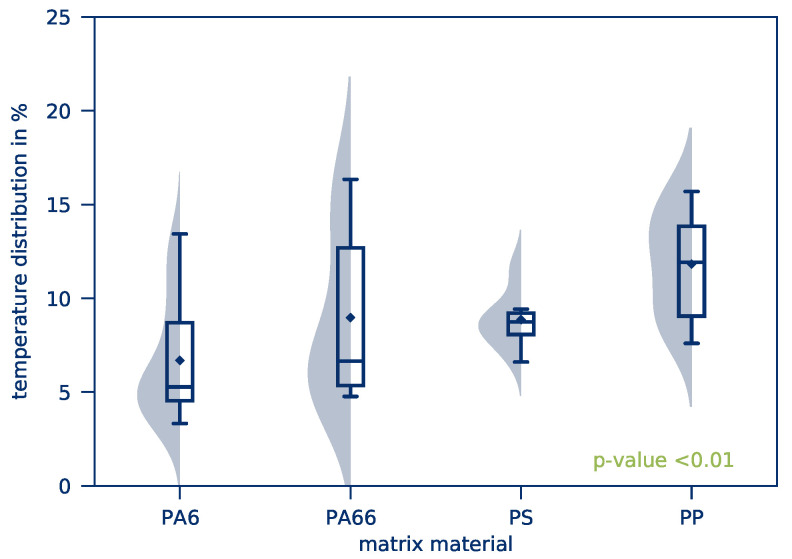
*Temperature distribution* as a function of the matrix material.

**Figure 16 polymers-12-02959-f016:**
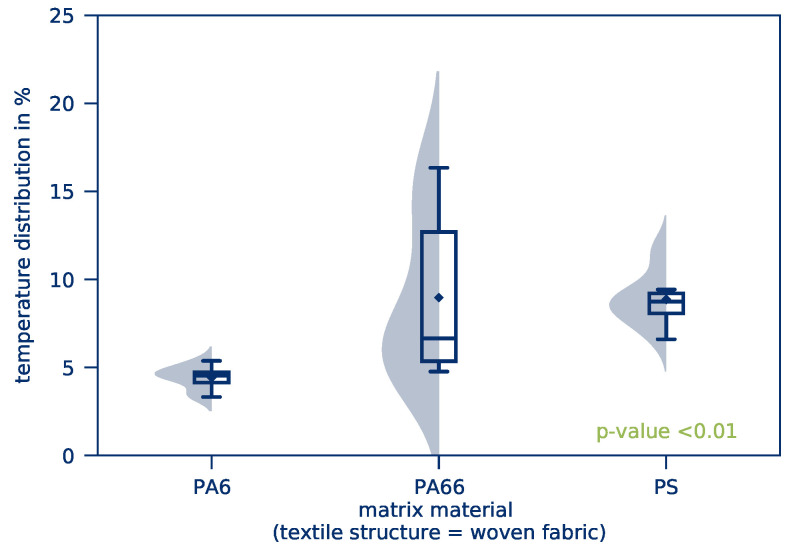
*Temperature distribution* as a function of the matrix material for woven fabric-based parts.

**Figure 17 polymers-12-02959-f017:**
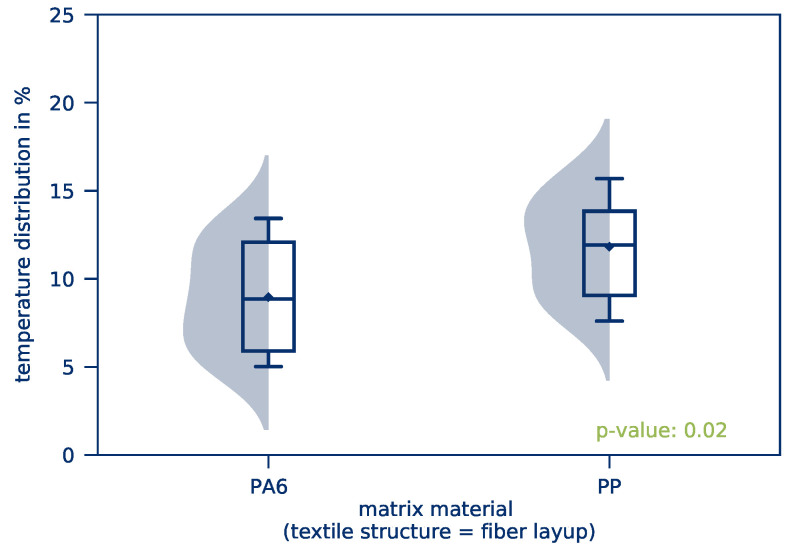
*Temperature distribution* as a function of matrix material for tape-based parts.

**Figure 18 polymers-12-02959-f018:**
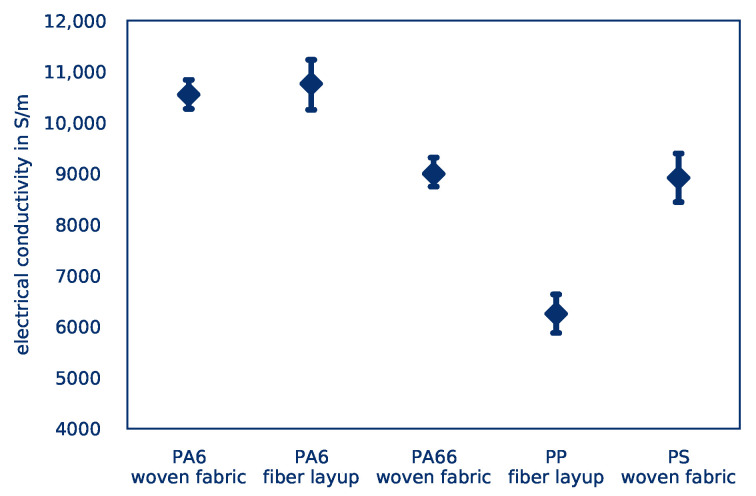
Electrical conductivity of the parts depending on matrix material.

**Figure 19 polymers-12-02959-f019:**
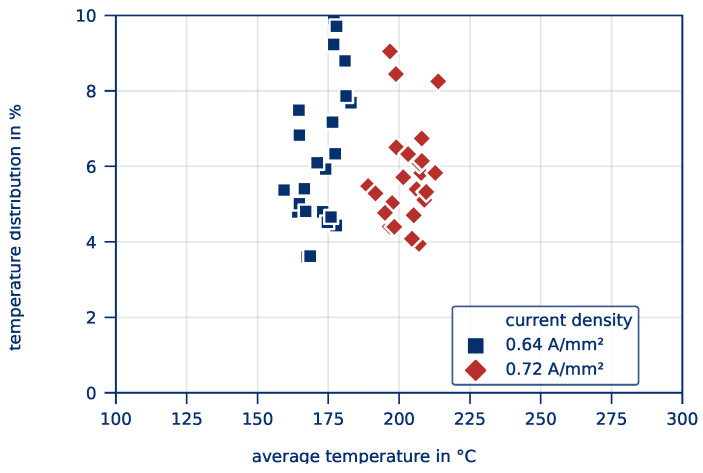
*Average part temperature* und *temperature distribution* as a result of varying currents for all experiments of the test series on the influence of the part’s size.

**Figure 20 polymers-12-02959-f020:**
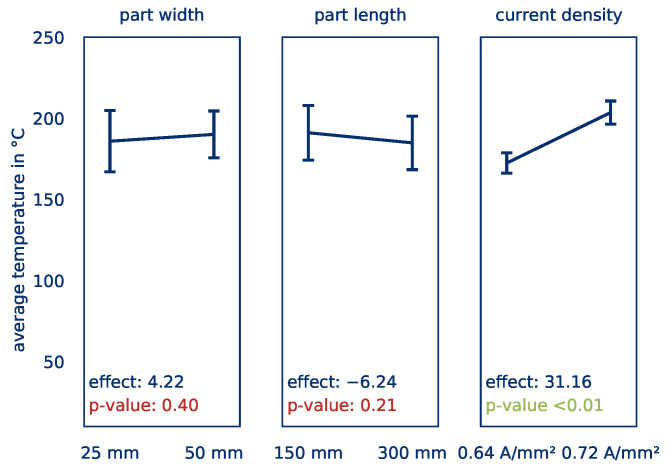
Calculated effects of the part width, part length, and current density on the *average part temperature*.

**Figure 21 polymers-12-02959-f021:**
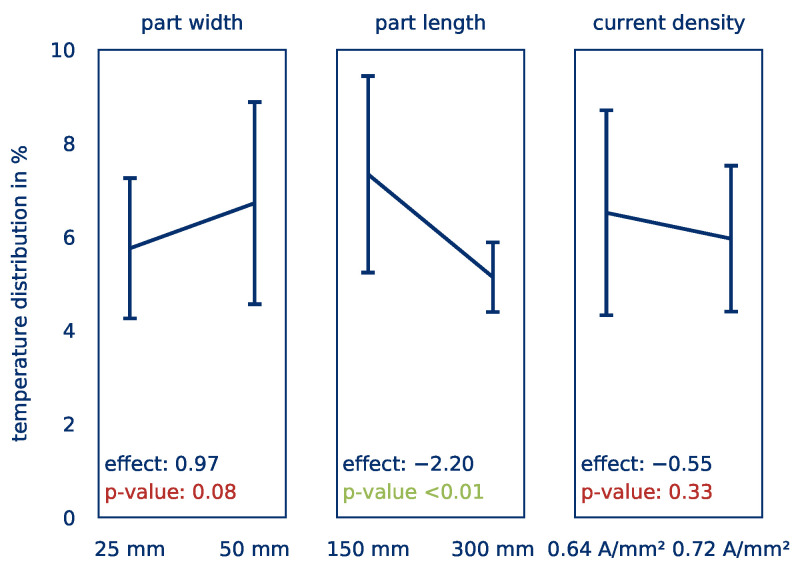
Calculated effects of the part width, part length, and current density on the *temperature distribution*.

**Table 1 polymers-12-02959-t001:** Investigation of the contacting: Parameters and parameter settings.

Factor	Setting −	Setting +
Current *I* in A	16	18
Contact force *F* in kN	10	20
Contact width *b* in mm	8	12
Contact shape *O*	flat	rounded

**Table 2 polymers-12-02959-t002:** Investigation of the materials: Parameters and parameter settings.

Producer	Textile Structure	Matrix Material
Bond Laminates	Woven fabric	PA66
Bond Laminates	Woven fabric	PA6
SGL Carbon	Tape based	PA6
SGL Carbon	Tape based	PP
Ineos Styrolution	Woven fabric	PS

**Table 3 polymers-12-02959-t003:** Investigation of the size: Parameters and parameter settings.

Factor	Setting −	Setting +
Current density *J* in A/mm	0.64	0.72
Part length *L* in mm	150	300
Part width *B* in mm	25	50

**Table 4 polymers-12-02959-t004:** Summary of the effects of the contacting on the heating behavior.

Target Value	Effect	Current	Contact Shape	Contact Width	Contact Force
*average part temperature*	$E_{\vartheta }$	32.73	2.36	−16.20	−2.04
*temperature distribution*	$E_{\sigma }$	−0.5	−0.35	−1.02	−0.1
*efficiency*	$E_{\varphi }$	0.02	0.01	0.04	0.03

**Table 5 polymers-12-02959-t005:** Summary of the effects of the part’s size on the heating behavior.

Target Value	Effect	Part Width	Part Length	Current Density
*average part temperature*	$E_{\vartheta }$	4.22	−6.24	31.16
*temperature distribution*	$E_{\sigma }$	0.97	−2.2	−0.55

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
