# Peer review of "Direct Joule Heating as a Means to Efficiently and Homogeneously Heat Thermoplastic Prepregs"

_polymers, 2020, doi:10.3390/polym12122959_

Round 1
Reviewer 1 Report
In the article “Direct Joule Heating as a Means to Efficiently and Homogeneously Heat Thermoplastic Prepregs” the authors describe the application of joule heating to thermoplastic prepregs before back-injection molding.
The experimental set-up is well described, the calculated effects of all studied parameters are clearly resumed in graphs and explained in the article.
Discussion and conclusions are supported by experimental evidences.
In the following lines, some minor corrections are requested:
Line 172: Please complete [?]
Line 257: Please indicate the Figure number.
Author Response
Dear Reviewer,
We would like to thank you very much for your review and your supportive comments. According to your specifications, we updated the missing reference and figure link.
Reviewer 2 Report
The paper entitled Direct Joule Heating as a Means to Efficiently and Homogeneously Heat Thermoplastic Prepregs is a very well written paper. I enjoyed reading it. The authors have explained well the importance of joule heating and other processes which will definitly help the new researchers to understand these phenomenon. However, the references are a bit older and it would be better to add some latest references from 2017-2020, if available on the subject.
Author Response
Dear Reviewer,
We would like to thank you very much for your review and your supportive comments. As per your request, we did a renewed literature research and added more recent papers to the reference list. To our knowledge, Joule heating of thermoplastic prepregs for back-injection molding has only been investigated by Hemmen and our group, which is why little research is published on that particular topic. Joule heating has been used for other applications however, which we included in the paper. The added references are highlighted in the text.
Reviewer 3 Report
The authors of this manuscript are reporting on direct Joule heating as a means to efficiently and homogeneously heat thermoplastic prepregs. I have enjoyed reading this paper. The work is interesting and seems to be original. But there are several shortcomings in the manuscript which must be addressed before publication in polymers. My comments to the authors are following:
Abstract:
The abstract is well written. However, incorporation of qualitative values of the finding of this study will be appreciated.
Introduction:
Introduction part has been divided into three sections. In the last section the authors are writing about setting of their hypothesis. The hypothesis is derived that the use of Joule heating of carbon fiber reinforced thermoplastics can be extended from resistance welding to heating of prepregs for back-injection molding, and that sufficiently fast and efficient heating can be achieved while maintaining a narrow temperature distribution.
However, it is not clear what they want to understand and what are prediction??
Novelty of the work should be emphasized more.
Experimental:
The % purity and molecular weight of polyamide etc should be included in the revised version.
The authors write that Joule heating occurs due to the movement of electrodes in a conductor. How does movement of electrodes possible in conductor?
There are five equations in the manuscript. However the parameters in equations 1-4 are not clearly defined.
Results
There are 21 figures in the manuscript. It will be more appropriate to shift some figures to supporting information.
Though the data are well presented, there is still a room for further improving the relevant discussion especially the statistical analysis. It is advised to compare the finding of this study with updated literature in the form of data presented in tabular form.
References:
Please update the references.
Author Response
Dear Reviewer,
We would like to thank you very much for your review and your supportive comments. Please find below a list of changes we made in accordance with your suggestions. Changes to the manuscript are marked in yellow.
Abstract:
"Incorporation of qualitative values of the finding of this study will be appreciated.":
The abstract length is limited to 200 words by the journal, which is why we kept it brief and did not include quantitative results. We believe that the inclusion of quantitative data would require a more elaborate description of the findings, resulting in a much longer abstract. If you still feel that a inclusion of quantitative data is required in the abstract, we are happy to restructure the abstract accordingly.
Hypothesis:
"However, it is not clear what they want to understand and what are prediction? Novelty of the work should be emphasized more."
Thank you very much for suggesting a clarification of our hypothesis. We believe that a clearly stated hypothesis is crucial for every paper and are happy to improve ours. We changed the text to further elaborate on our hypothesis and assumptions. This should also emphasize the novelty of our research.
Experimental:
"The % purity and molecular weight of polyamide etc. should be included in the revised version.":
We did not fully analyze the materials used in this study, but we added a link to the material data-sheet in the references. Since we used industrial grade materials, not much information is available on the polymers used. We did, however, measure the electrical conductivity, which is part of the results section and used specific heat capacities from literature.
"The authors write that Joule heating occurs due to the movement of electrodes in a conductor. How does movement of electrodes possible in conductor?":
We do not really understand what the suggested changes to the manuscript are. Joule heating occurs because of electrons accelerated by an electric potential in dependence of the material’s electrical resistance. This is true for any given material/electrical conductor and is the basis of Joule heating. Would you be willing to further elaborate on your comment?
"There are five equations in the manuscript. However the parameters in equations 1-4 are not clearly defined.":
Thank you for your comment. We updated the equation descriptions.
Results
"It will be more appropriate to shift some figures to supporting information.":
We changed the positioning of figures to better suit the manuscript.
"Though the data are well presented, there is still a room for further improving the relevant discussion especially the statistical analysis.":
Thank you for your comment. We revised the results section with an updated statistical analysis and tried to further improve readability and understanding.
"It is advised to compare the finding of this study with updated literature in the form of data presented in tabular form.":
We added a summary of our findings in tabular form at the summary of the sections on the influence of contacting and part size. To our knowledge, no other group except Hemmen's did research on the direct Joule Heating of thermoplastic prepregs, which is why a comparison to literature is not possible (yet).
References
"Please update the references.":
We included more recent references. To our knowledge, Joule heating of thermoplastic prepregs for back-injection molding has only been investigated by Hemmen and our group, which is why little research is published on that particular topic. Joule heating has been used for other applications however, which we included in the paper. The added references are highlighted in the text.